# A state-space model for inferring effective connectivity of latent neural dynamics from simultaneous EEG/fMRI

**Tao Tu**
Columbia University
tt2531@columbia.edu

**John Paisley**
Columbia University
jpaisley@columbia.edu

**Stefan Haufe**
Charité – Universitätsmedizin Berlin
stefan.haufe@charite.de

**Paul Sajda**
Columbia University
psajda@columbia.edu

## Abstract

Inferring effective connectivity between spatially segregated brain regions is important for understanding human brain dynamics in health and disease. Non-invasive neuroimaging modalities, such as electroencephalography (EEG) and functional magnetic resonance imaging (fMRI), are often used to make measurements and infer connectivity. However most studies do not consider integrating the two modalities even though each is an indirect measure of the latent neural dynamics and each has its own spatial and/or temporal limitations. In this study, we develop a linear state-space model to infer the effective connectivity in a distributed brain network based on simultaneously recorded EEG and fMRI data. Our method first identifies task-dependent and subject-dependent regions of interest (ROI) based on the analysis of fMRI data. Directed influences between the latent neural states at these ROIs are then modeled as a multivariate autogressive (MVAR) process driven by various exogenous inputs. The latent neural dynamics give rise to the observed scalp EEG measurements via a biophysically informed linear EEG forward model. We use a mean-field variational Bayesian approach to infer the posterior distribution of latent states and model parameters. The performance of the model was evaluated on two sets of simulations. Our results emphasize the importance of obtaining accurate spatial localization of ROIs from fMRI. Finally, we applied the model to simultaneously recorded EEG-fMRI data from 10 subjects during a Face-Car-House visual categorization task and compared the change in connectivity induced by different stimulus categories.

## 1 Introduction

Identifying the spatiotemporal dependence among distributed cortical regions is often seen as crucial for understanding the macro-scale neural dynamics underlying human cognition. Such spatiotemporal dependencies can be quantified statistically by the modeling of effective connectivity, which is defined as the time-lagged influence of one brain region over another [1]. Effective connectivity has been introduced in the framework of dynamic causal modeling (DCM). DCM uses a state-space model with hidden state variables to describe task-dependent "causal" interactions between latent neural states and how the activity of regional neural states translates into observed neural measurements [2, 3]. Estimating effective connectivity between anatomically segregated brain regions is a challenging problem for several reasons: 1) the inference is made on unobserved latent states rather than directly on the observations; 2) latent neural dynamics often evolve on a fast time scale so it requires the

observation time series to be measured on a similar temporal scale; 3) accurate spatial localization of the activated brain regions is often a prerequisite for the specification of a meaningful dynamic causal model.

To address these challenges, a number of state-space based modeling techniques have been developed and applied to a variety of non-invasive neuroimage modalities such as electroencephalography (EEG)/ magnetoencephalography (MEG), functional magnetic resonance imaging (fMRI), and functional near-infrared spectroscopy (fNIRS). Modalities like EEG and MEG with high temporal resolution offer advantages in terms of measuring and inferring effective connectivity. Cheung et al. [4] proposed a state-space model where the latent neural dynamics at pre-defined ROIs were modeled as a multivariate autoregressive (MVAR) process. They assumed a known EEG forward model with unknown spatial distribution of the EEG sources within each ROI. Haufe et al. [5] used a similar MVAR approach to model the connectivity in EEG source space where the spatial source demixing was optimized jointly with the connectivity estimation. David et al. [6] used a nonlinear hierarchical neural mass model for the "casual" modeling of evoked responses in EEG/MEG. Another model for evoked responses in MEG/EEG was proposed by Yang et al. [7] where a time-varying MVAR model was used to estimate the dynamic connectivity among multiple ROIs. In contrast to the work by Cheung et al. [4], they also used a known MEG/EEG forward model for the evoked responses, but sources within the same ROI were modeled as independent Gaussian variables. Other dynamical models leveraging the relatively high spatial resolution of fMRI [3, 8, 9] and fNIRs [10, 11] have also been developed for brain connectivity analysis.

All of these inference methods are based on neural measurements from a single modality, and therefore suffer from potentially suboptimal estimates of the true latent neural dynamics due to the limitation in spatial or temporal resolution of the modality. Simultaneous EEG-fMRI is a neuroimaging technique that leverages the complementary strengths of both modalities, namely 3D spatial resolution of fMRI and temporal resolution of EEG. Given that the data from two modalities are recorded under identical experimental conditions, one can use fMRI activations as a spatial prior to improve the accuracy of EEG source localization [12, 13, 14].

In this paper we propose a linear state-space model for estimating the effective connectivity using, as observations, data from simultaneously recorded EEG and fMRI. Our goal is to combine EEG with fMRI to arrive at estimates of the latent neural dynamics with high spatiotemporal resolution. Since fMRI offers significant advantage over EEG in terms of spatially localizing potential source activity, we first identify task-specific ROIs from the analysis of fMRI data on each individual subject. The locations of these ROIs are used as spatial constraints to inform the effective connectivity modeling of EEG. Similar to the ROI source model proposed by Yang et al. [7], we also model the latent state variables as the mean source activity at each ROI. Each source inside one ROI follows a Gaussian distribution with the ROI mean and a shared unknown variance parameter. In contrast to [7], we model the state equation as an MVAR process, which describes the directed interactions between latent states driven by deterministic inputs specific to an experiment. Inputs can directly influence the activity at a particular region (external input) or they can modulate the connectivity between regions. Finally, an EEG forward model based on a pre-estimated lead field matrix was constructed together with the ROI source model to generate scalp EEG observations. We use a mean-field variational Bayesian approach to infer the posterior distribution of latent variables and model parameters. The posterior estimates are updated efficiently via a sequential Kalman filter and the use of conjugate priors. We evaluated the model performance on two sets of simulations and demonstrated the importance of the spatial specificity provided by fMRI. We then applied the state-space model to simultaneously EEG-fMRI recordings from 10 subjects during a face-car-house rapid decision-making task.

## 2   Model

**Model description**   Our linear state-space model for inferring the latent neural dynamics consists of a state equation and two observation equations for EEG. In the state equation, we model the temporal dependence between latent state variables as a first-order MVAR process in the presence of external and context-dependent inputs:

$$\mathbf{s}_t = \mathbf{A}\mathbf{s}_{t-1} + \sum_{k=1}^{K} \mathbf{B}^k m_t^k \mathbf{s}_{t-1} + \mathbf{D}\mathbf{u}_t + \boldsymbol{\omega}_t \qquad (1)$$

where $\mathbf{s}_t$ is an $S \times 1$ vector of latent state variables at time $t$. Each element in $\mathbf{s}_t$ represents the mean activity of all EEG sources within one of $S$ ROIs. $\mathbf{A}$ is an $S \times S$ intrinsic connectivity matrix wherein each entry denotes the connection strength between a pair of latent variables in the absence of input. $m_t^k$, where $k = 1, 2, ...K$, represents the $k^{th}$ modulatory input at time $t$. $\mathbf{B}^k \in \mathbb{R}^{S \times S}$ is the $k^{th}$ modulatory connectivity matrix where each element denotes the change in connectivity induced by the modulatory input $m_t^k$. $\mathbf{u}_t$ is an $S \times 1$ vector that denotes the external input at each ROI and $\mathbf{D}$ is an $S \times S$ diagonal matrix whose diagonal element denotes the strength of $\mathbf{u}_t$. $\boldsymbol{\omega}_t \in \mathbb{R}^{S \times 1}$ is a Gaussian state noise vector at time $t$ with a zero mean and a diagonal covariance matrix $\mathbf{Q}_s$. This bilinear model used to approximate the latent state dynamics modulated by task demand is similar to that in [3].

The observation model for EEG consists of two equations. We used a volumetric source model which assumes that EEG sources are uniformly distributed on a 3-D grid inside the brain. The position and the orientation of each source (dipole) is fixed and pre-estimated from real data in this model. Source activity $\mathbf{x}_t$ propagates through brain tissues and generates EEG potentials $\mathbf{y}_t$ measured by electrodes placed on the scalp via the following linear forward model:

$$\mathbf{y}_t = \mathbf{L}\mathbf{x}_t + \mathbf{e}_t, \tag{2}$$

where $\mathbf{y}_t \in \mathbb{R}^{M \times 1}$ is the EEG observations measured from $M$ channels at time $t$. $\mathbf{x}_t \in \mathbb{R}^{U \times 1}$ is the activity of $U$ EEG sources at time $t$. $\mathbf{L} \in \mathbb{R}^{M \times U}$ is the lead field matrix that describes the mapping from EEG source space to channel space. $\mathbf{L}$ in our model was pre-computed by solving the EEG forward modeling problem [15]. $\mathbf{e}_t$ is an $M \times 1$ vector that models the noise at each channel as a Gaussian with zero mean and covariance matrix $\mathbf{Q}_y$. Solving $\mathbf{x}_t$ from $\mathbf{y}_t$ is called EEG inverse modeling or source localization. It is an ill-posed problem since $U \gg M$. Many EEG source localization methods such as minimum-norm estimation (MNE) [16] require the estimate of $\mathbf{Q}_y$ from baseline data. The solution to EEG source localization is not unique and often not robust, especially based on EEG data alone. Therefore, we did not model each single source in the whole brain as a latent variable. Similar as in [7], the source activity $\mathbf{x}_t$ in our model is composed of the latent variables $\mathbf{s}_t$ by the following equation:

$$\mathbf{x}_t = \mathbf{G}\mathbf{s}_t + \boldsymbol{\epsilon}_t \tag{3}$$

where $\mathbf{G} \in \mathbb{R}^{U \times S}$ is a binary indicator matrix. Each row of $\mathbf{G}$ is a one-hot vector that encodes the membership of each source in one of the ROIs. $\boldsymbol{\epsilon}_t \in \mathbb{R}^{U \times 1}$ is a Gaussian noise term with zero mean and $U \times U$ diagonal covariance matrix $\mathbf{Q}_x$. Consequently, each source in $\mathbf{x}_t$ is Gaussian distributed around its corresponding ROI mean in $\mathbf{s}_t$ with a variance $\sigma_r^2, r = 1, \ldots, S$ specified in the diagonal elements of $\mathbf{Q}_x$. If a source in $\mathbf{x}_t$ is not contained in any ROI, it is modeled as a Gaussian variable with a zero mean and variance $\sigma_0^2$. This model assumes that all sources in the $r^{th}$ ROI have the same variance parameter $\sigma_r^2$, while sources that do not belong to any ROI have the same variance parameter $\sigma_0^2$. Therefore, there are only $S + 1$ distinct elements in the diagonal of $\mathbf{Q}_x$.

Substituting (3) into (2) and eliminating $\mathbf{x}_t$, the EEG observation model can be expressed as:

$$\mathbf{y}_t = \mathbf{C}\mathbf{s}_t + \boldsymbol{\phi}_t \tag{4}$$

where $\mathbf{C} = \mathbf{L}\mathbf{G}$ is a known $M \times S$ matrix and $\boldsymbol{\phi}_t$ is the Gaussian noise term at time $t$ with a zero mean and an $M \times M$ covariance matrix $\mathbf{R} = \mathbf{Q}_y + \mathbf{L}\mathbf{Q}_x\mathbf{L}'$. $\mathbf{L}'$ denotes the transpose of $\mathbf{L}$.

Taken together, our linear state-space model can be expressed as (see Figure 1 for illustration):

$$\mathbf{s}_t|\mathbf{s}_{t-1} \sim \mathcal{N}(\mathbf{A}\mathbf{s}_{t-1} + \sum_{k=1}^{K} \mathbf{B}^k m_t^k \mathbf{s}_{t-1} + \mathbf{D}\mathbf{u}_t, \mathbf{Q}_s), \qquad \mathbf{y}_t|\mathbf{s}_t \sim \mathcal{N}(\mathbf{C}\mathbf{s}_t, \mathbf{R}). \tag{5}$$

**Model inference**  Given EEG observations $\mathbf{Y} = \{\mathbf{y}_t\}_{t=1}^{T}$, we use the mean-field variational Bayesian (VB) approximation to make inference on the posterior distributions of the latent state variables $\mathbf{S} = \{\mathbf{s}_t\}_{t=1}^{T}$ and the unknown model parameters $\boldsymbol{\theta} = \{\mathbf{A}, \{\mathbf{B}_k\}_{k=1}^{K}, \mathbf{D}, \mathbf{Q}_s, \mathbf{R}\}$. Figure 1B shows the probabilistic graphical representation of our model. In VB, we make analytical approximation to the joint posterior distribution $p(\mathbf{S}, \boldsymbol{\theta}|\mathbf{Y})$ in order to maximize the evidence lower bound (ELBO)[17]:

$$\mathcal{L}(q) = \int q(\mathbf{S}, \boldsymbol{\theta}) \log \frac{p(\mathbf{S}, \boldsymbol{\theta}, \mathbf{Y})}{q(\mathbf{S}, \boldsymbol{\theta})} d\boldsymbol{\theta} d\mathbf{S} \tag{6}$$

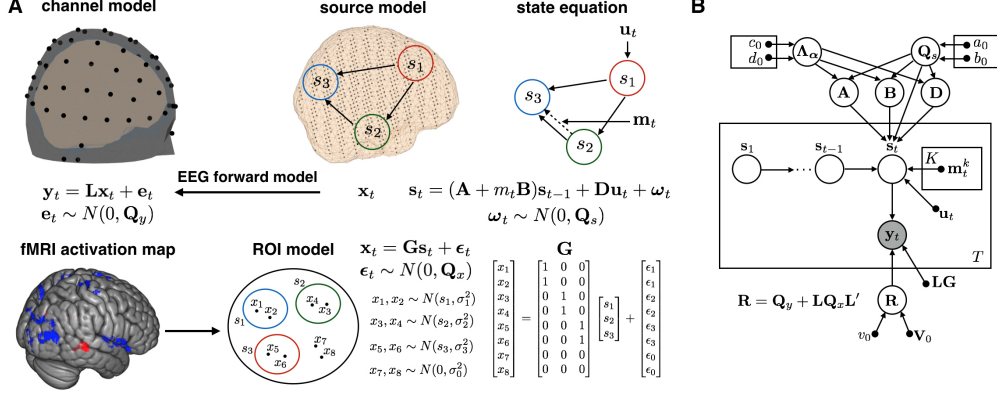

Figure 1: Model overview. A, Illustration of the linear state-space model for simultaneous EEG/fMRI. B, Probabilistic graphical representation of the model.

where $q(\mathbf{S}, \boldsymbol{\theta})$ is an arbitrary density from a family of variational distributions. It is easy to show that the ELBO objective is maximized when $q(\mathbf{S}, \boldsymbol{\theta}) = p(\mathbf{S}, \boldsymbol{\theta}|\mathbf{Y})$. Since $p(\mathbf{S}, \boldsymbol{\theta}|\mathbf{Y})$ is often intractable, we choose a density from the mean-field variational family having the form $q(\mathbf{S}, \boldsymbol{\theta}) = q(\mathbf{S}|\mathbf{Y})q(\boldsymbol{\theta}|\mathbf{Y})$ to approximate $p(\mathbf{S}, \boldsymbol{\theta}|\mathbf{Y})$. The solution that maximizes $\mathcal{L}(q)$ satisfies [17]:

$$\log q(\mathbf{S}|\mathbf{Y}) \propto \mathbb{E}_{\boldsymbol{\theta}}(\log p(\mathbf{S}, \boldsymbol{\theta}, \mathbf{Y})) \tag{7}$$

$$\log q(\boldsymbol{\theta}|\mathbf{Y}) \propto \mathbb{E}_{\mathbf{S}}(\log p(\mathbf{S}, \boldsymbol{\theta}, \mathbf{Y})) \tag{8}$$

where the expectation is taken with respect to $q(\boldsymbol{\theta}|\mathbf{Y})$ and $q(\mathbf{S}|\mathbf{Y})$ respectively.

Equation (7) is the VB-E step where we estimate the posterior distribution of latent variable $q(\mathbf{S}|\mathbf{Y})$ given the current estimate of $q(\boldsymbol{\theta}|\mathbf{Y})$. Since we assume Gaussian posterior on $\mathbf{S}$, we use Kalman filtering and smoothing to sequentially update the posterior mean $\boldsymbol{\mu}_t^T$ and covariance $\boldsymbol{\Sigma}_t^T$ of the latent variables at every time $t$. More details of the derivation are provided in Appendix.

Equation (8) is the VB-M step where we update the posterior distribution of model parameters $q(\boldsymbol{\theta}|\mathbf{Y})$ given the current estimate of $q(\mathbf{S}|\mathbf{Y})$. For the state model parameters $\boldsymbol{\theta}^S = \left\{\mathbf{A}, \{\mathbf{B}_k\}_{k=1}^K, \mathbf{D}, \mathbf{Q}_s\right\}$, we choose a Gaussian-Gamma conjugate prior according to the principle of automatic relevance determination (ARD). ARD assigns a separate shrinkage prior to each element of the connectivity matrices which in turn is adjusted by a hyper-prior [17, 18]. It encourages a sparse structure in the connectivity matrices to enhance interpretability. The use of conjugate priors also allows one to obtain closed-form solution for the posterior updates of model parameters. Since we assume the state noise covariance $\mathbf{Q}_s$ to be diagonal, we can estimate each row in the model parameters $\boldsymbol{\theta}^S$ separately. Specifically, the $r^{th}$ row of the state equation can be expressed as:

$$s_t[r] = \boldsymbol{\eta}'[r]\tilde{\mathbf{s}}_t[r] + \omega_t[r], \quad \omega_t[r] \sim \mathcal{N}(0, \beta^{-1}[r]), \quad \beta[r] = 1/\mathbf{Q}_s(r, r) \tag{9}$$

where $\tilde{\mathbf{s}}_t[r] = \begin{bmatrix} \tilde{\mathbf{F}}_t \mathbf{s}_{t-1} \\ u_t[r] \end{bmatrix}$, $\tilde{\mathbf{F}}_t = \begin{bmatrix} \mathbf{I}_S & m_t^1 \mathbf{I}_S \dots & m_t^K \mathbf{I}_S \end{bmatrix}'$ and $\boldsymbol{\eta}[r] = [\mathbf{a}[r], \mathbf{b}_1[r], ..., \mathbf{b}_K[r], d[r]]'$. $\beta[r]$ is the precision of the state noise at the $r^{th}$ row; $\mathbf{a}[r]$ and $\mathbf{b}^k[r]$ are the $r^{th}$ rows of $\mathbf{A}$ and $\mathbf{B}_k$, respectively; $d[r]$ is the $r^{th}$ diagonal element of $\mathbf{D}$.

We assume the following Gaussian-Gamma conjugate priors for $\boldsymbol{\eta}[r]$, $\beta[r]$, and $\boldsymbol{\alpha}$ [19]:

$$p(\boldsymbol{\eta}[r], \beta[r]|\boldsymbol{\alpha}) = \mathcal{N}\left(0, (\beta[r]\boldsymbol{\Lambda}_{\boldsymbol{\alpha}})^{-1}\right) \text{Gamma}(a_0, b_0), \quad p(\boldsymbol{\alpha}) = \prod_{i=1}^{(K+1)S+1} \text{Gamma}(c_0, d_0) \tag{10}$$

where $\boldsymbol{\alpha} = [\alpha_1, \alpha_2, ..., \alpha_{(K+1)S+1}]$ is a vector of hyperparameters on each element of $\boldsymbol{\eta}[r]$ and $\boldsymbol{\Lambda}_{\boldsymbol{\alpha}}$ is a diagonal matrix with the vector $\boldsymbol{\alpha}$. Each hyperparameter in $\boldsymbol{\alpha}$ has a separate Gamma prior. The variational joint posterior for $\boldsymbol{\eta}[r]$ and $\beta[r]$ has the same form as their priors:

$$q(\boldsymbol{\eta}[r], \beta[r]|\mathbf{Y}) = \mathcal{N}(\bar{\boldsymbol{\mu}}[r], \beta^{-1}[r]\bar{\boldsymbol{\Sigma}}[r])\text{Gamma}(\bar{a}[r], \bar{b}[r]) \tag{11}$$

where

$$\bar{\Sigma}^{-1}[r] = \begin{bmatrix} \sum_{t=2}^{T} \tilde{\mathbf{F}}_t \mathbb{E}_{\mathbf{s}}[\mathbf{s}_{t-1}\mathbf{s}'_{t-1}]\tilde{\mathbf{F}}'_t & \sum_{t=2}^{T} \tilde{\mathbf{F}}_t \boldsymbol{\mu}^T_{t-1} u_t[r] \\ \sum_{t=2}^{T} u_t[r](\boldsymbol{\mu}^T_{t-1})'\tilde{\mathbf{F}}'_t & \sum_{t=2}^{T} (u_t[r])^2 \end{bmatrix} + \mathbb{E}_{\boldsymbol{\alpha}}(\Lambda_{\boldsymbol{\alpha}}) \tag{12}$$

$$\bar{\boldsymbol{\mu}}[r] = \bar{\Sigma}[r]\begin{bmatrix} \sum_{t=2}^{T} \tilde{\mathbf{F}}_t \mathbb{E}_{\mathbf{s}}[s_t[r]\mathbf{s}_{t-1}] \\ \sum_{t=2}^{T} u_t[r]\mu^T_t[r] \end{bmatrix}, \quad \mathbb{E}_{\boldsymbol{\alpha}}(\Lambda_{\alpha}) = \operatorname{diag}\left(\frac{\bar{c}_1}{\bar{d}_1}, \frac{\bar{c}_2}{\bar{d}_2}, ..., \frac{\bar{c}_{(K+1)S+1}}{\bar{d}_{(K+1)S+1}}\right) \tag{13}$$

$$\bar{a}[r] = a_0 + \frac{T-1}{2}, \quad \bar{b}[r] = b_0 + \frac{1}{2}\left[\sum_{t=2}^{T} \mathbb{E}_{\mathbf{s}}[(s_t[r])^2] - \bar{\boldsymbol{\mu}}'[r]\bar{\Sigma}^{-1}[r]\bar{\boldsymbol{\mu}}[r]\right] \tag{14}$$

The posterior for each hyperparameter $\alpha_j, j = 1, 2, ..., (K+1)S+1$ can be computed independently:

$$q(\alpha_j|\mathbf{Y}) = \operatorname{Gamma}(\alpha_j|\bar{c}_j, \bar{d}_j) \tag{15}$$

where

$$\bar{c}_j = c_0 + \frac{1}{2}, \quad \bar{d}_j = d_0 + \frac{1}{2}\left[\frac{\bar{a}[r]}{\bar{b}[r]}(\bar{\boldsymbol{\mu}}[r,j])^2 + \bar{\Sigma}_r[j,j]\right] \tag{16}$$

$\bar{\boldsymbol{\mu}}[r,j]$ is the $j^{th}$ element of $\bar{\boldsymbol{\mu}}[r]$ and $\bar{\Sigma}_r[j,j]$ is the $j^{th}$ diagonal element of $\bar{\Sigma}[r]$.

The noise covariance $\mathbf{R}$ comprises two unknown quantities $\mathbf{Q}_y$ and $\mathbf{Q}_x$. Choosing a conjugate prior for each of them individually is difficult. Since $\mathbf{Q}_y$ and $\mathbf{Q}_x$ are not of primary interest in our study, we optimize $\mathbf{R}$ directly. We set the inverse Wishart prior $IW(v_0, \mathbf{V}_0)$ on $\mathbf{R}$ [20]:

$$q(\mathbf{R}|\mathbf{y}) = IW(v_n, \mathbf{V}_n) \tag{17}$$

where

$$v_n = v_0 + T, \quad \mathbf{V}_n = \mathbf{V}_0 + \left(\sum_{t=1}^{T}(\mathbf{y}_t - \mathbf{C}\boldsymbol{\mu}^T_t)(\mathbf{y}_t - \mathbf{C}\boldsymbol{\mu}^T_t)' + \mathbf{C}\Sigma^T_t\mathbf{C}'\right) \tag{18}$$

The implementation of the algorithm in Matlab and the dataset are available at `https://github.com/taotu/VBLDS_Connectivity_EEG_fMRI`.

## 3 Results

We first evaluated the performance of the state-space model on simulated datasets and then applied the model to real simultaneously recorded EEG and fMRI data (see more details in Appendix). In the simulation study, we assessed the performance of the model when spatial localization of ROIs is inaccurate, simulating the scenario when fMRI information is not available. We generated two simulation scenarios corresponding to two different types of EEG-fMRI experiment designs: a block design and an event-related design. For analysis of the real simultaneous EEG-fMRI data, we applied the state-space model on the EEG data recorded simultaneously with fMRI to infer the induced connectivity change between brain regions activated during a Face-Car-House visual categorization task. Combining the subject-specific fMRI activation maps and the EEG temporal dynamics enabled us to compare differences in modulatory connectivity induced by face stimuli vs. house stimuli.

### 3.1 Simulations

**Scenario 1: Block Design** We simulated the latent dynamics in the brain network consisting of $S = 5$ ROIs using the structure shown in Figure 2A. The external input was modeled as a sequence of impulse functions with an inter-stimulus interval (ISI) uniformly drawn between 2 s to 2.5 s (longer than the fMRI repetition time TR=2 s). The modulatory input was modeled as alternating on-off blocks with a block duration of 20 s to simulate a block design fMRI experiment where change in the network connectivity could be induced by stimulus presentation or alternation of cognitive states (such as attention and salience). The external input feeds into FFA with a strength of 0.9 and the modulatory input changes the connection strength from SPL to PPA and from ACC to FEF. In particular, the direction of the modulatory connection from SPL to PPA is opposite to that of the intrinsic connection between them. The state covariance $\mathbf{Q}_s$ was set to be the identity matrix. The ROI variance $\sigma_r^2, r = 0, 1, \ldots, S$ in $\mathbf{Q}_x$ was drawn from a Gamma distribution $\boldsymbol{\Gamma}(0.2, 1)$ whose shape and scale parameters were estimated from real data. The EEG measurement noise covariance $\mathbf{Q}_y$ was also estimated from real data during the baseline period. We simulated the latent ROI mean

activity and EEG data for a duration of $T = 8$ min with a sampling rate of 100 Hz. The unit of the simulated EEG measurements was microvolt.

**Scenario 2: Event-related Design** To mimic a more realistic EEG-fMRI experiment design, we simulated two modulatory inputs that induce different connectivity patterns shown in Figure 2B. The modulatory inputs were modeled as a sequence of discrete events with a duration of 2 s. The ISI was also drawn uniformly from 2 s to 2.5 s so that there was no overlap between the two inputs. Other parameters were the same as in scenario 1. The aim of this simulation was to test whether the algorithm could correctly distinguish the modulatory connectivity matrices induced by different modulatory inputs.

To illustrate the value of the high spatial specificity provided by fMRI, we simulated an 'EEG-only' condition where fMRI data was not available. To achieve this, the anatomical region of each ROI was dilated so that the number of sources erroneously included was approximately 30% of the total number of true sources across 5 ROIs. The direct outcome of this spatial smearing was that more rows in $\mathbf{G}$ would have nonzero entries. In the absence of fMRI data, one typically has to define ROIs based on atlases defined based on structural brain images, which may cause inaccurate spatial localization of ROIs. We compared the performance of the algorithm between the 'EEG-fMRI' and the 'EEG-only' conditions to highlight the importance of the spatial resolution added by fMRI.

Ten independent simulation datasets were generated for each scenario. For each simulation dataset, we applied the EEG-fMRI method and the EEG-only method separately. Since we simulated relatively large samples, we chose small non-informative priors for the model parameters. Two methods were initialized with the same set of parameters (see Appendix). The performance of the algorithm in recovering the intrinsic and modulatory connectivity matrices $\mathbf{A}$ and $\mathbf{B}$ as well as the noise covariance matrices $\mathbf{Q}_s$ and $\mathbf{R}$ was evaluated using the relative error between the true and estimated values defined as:

$$e = \frac{||\mathbf{X}_{true} - \hat{\mathbf{X}}_{est}||_F}{||\mathbf{X}_{true}||_F} \tag{19}$$

where $|| \cdot ||_F$ is the Frobenius norm of a matrix. Statistical inference on the entries of the connectivity matrices is straightforward since they have Gaussian posteriors. Prior to calculating the relative error of $\mathbf{A}$ and $\mathbf{B}$, we thresholded each connection according to its posterior distribution at $P < 0.05$ with Bonferroni correction to account for multiple comparison (N=108). Figure 2 shows the comparison of the relative error between the two methods. For both simulation scenarios, EEG-fMRI method generated more accurate estimation than EEG-only method. In our EEG-only simulation, even though only a small number of sources (38) that did not contribute to the underlying dynamics was falsely assigned to all ROIs, the performance largely decreased. In practice, without the fMRI data, one would only get more inaccurate spatial localization of ROIs.

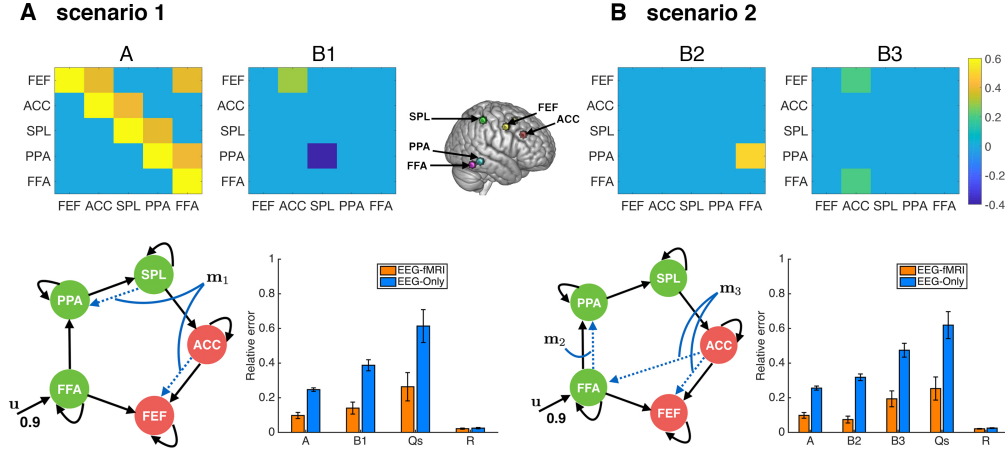

Figure 2: Performance on two sets of simulations. A, 5-node network structure in scenario 1 and the relative error of $\mathbf{A}$, $\mathbf{B1}$, $\mathbf{Q}_s$, $\mathbf{R}$ for the EEG-fMRI (orange) and the EEG-only (blue) conditions. B, Similar comparison for scenario 2 where two modulatory matrices $\mathbf{B2}$ and $\mathbf{B3}$ were simulated. Blue line denotes intrinsic connection and blue dotted line denotes modulatory connection between two nodes. Error bar represents the standard error of the mean across 10 independent simulations.

## 3.2 Simultaneous EEG-fMRI Data

We then applied our state-space model method on simultaneously recorded EEG and fMRI data from 10 subjects. The data were recorded when subjects performed an event-related three-choice visual categorization task. On each trial, an image of a face, car, or house was presented at random for 100 ms. The ISI ranged uniformly between 2 s and 2.5 s. Subjects reported their choice of the image category by pressing one of the three buttons on an MR-compatible button response pad. Each subject completed 4 runs of the categorization task. In each run, there were 180 trials (60 per category) with a total duration of 560 s. Previous studies [21, 22] have implicated two spatially and temporally separate brain networks (which we term the 'early' and 'late' networks) during this rapid perceptual decision task based on an EEG-informed fMRI analysis approach. However, the latent brain dynamics were inferred from the fMRI data, which fluctuate on a much slower timescale than the latent neural processes. In this study, we leveraged the high temporal resolution of the EEG data in combination with the high spatial specificity of fMRI to estimate the latent brain dynamics underlying behavior in this task. In particular, we selected 3 regions (FFA, PPA, SPL) from the early network and 3 regions (ACC, premotor cortex, FEF) from the late network that constituent a brain network of 6 ROIs (Figure 3A). We added premotor cortex (PMC) in our analysis because the task involved motor planning and execution. FFA and PPA were determined based on a separate functional localizer task for each subject and they were included because of their selectivity in the early sensory processing of faces and houses. SPL, ACC and FEF have all been shown to involve at different stages of the perceptual decision-making. The ROIs were determined based on a group-level EEG-informed fMRI analysis in the standard space but were then transformed back into each subject's native anatomical space.

**Statistical inference** For each subject, we fitted the model to each of the 4 runs separately. The estimated modulatory connectivity matrices corresponding to face and house were z-scored and thresholded according to their posterior probability. We then performed a two-tailed z-test on the mean z-scored connectivity values across 40 runs from 10 subjects. Significance was determined at $p < 0.05$ with Bonferroni correction to account for multiple comparisons across three connectivity matrices. Significant differences between face and house networks was determined using a paired t-test on the z-scored connections at $p < 0.05$. Figure 3B shows the mean network connectivity pattern for face and house stimuli, respectively. Since both positive connections and negative connections are meaningful, we showed the absolute value of all significant connections.

We consider the effective connectivity we infer with respect to differences between face stimuli and house stimuli. Faces and houses are object types often used in fMRI and EEG studies to study object recognition and decision-making. Each of these stimulus categories is known to preferentially activate different regions of the brain (FFA for faces and PPA for houses/places). These stimuli are also interesting in this context since they are selected so that the organization of the features making up the objects overlaps (eyes and windows in same relative positions as are nose and door) and thus can be challenging to discriminate in the presence of visual noise and rapid stimulus presentation. Our results show that effective connectivity differences are apparent, specifically we see an increase in effective connectivity when a house is presented relative to when the stimulus is a face. The specific connections contributing to this difference are shown in Figure 3C. Interestingly, these differences involve connections with the ACC as well as the FEF and FFA, which are areas implicated in cognitive control, decision monitoring, attention and object recognition, especially of faces. The fact that the connections are more engaged for house stimuli suggests that there is more of a need to link these areas when a house is presented relative to a face–i.e this additional connectivity is required for recognizing a house relative to a face. Previous work [21] showed that network connectivity is likely a source of how bias effects toward faces are manifested in our choices. This current, though preliminary result, suggests that overcoming this bias requires additional network connectivity.

## 4 Discussion

Leveraging the complementary strengths of EEG and fMRI, we proposed a linear state-space model to estimate the effective connectivity between spatially segregated but functionally integrated brain regions. Specifically, we focused on the analysis of effective connectivity driven by various context-dependent inputs. We modeled the latent state variables as the mean source activity in each ROI and assumed that all source points belonging to one ROI are independent Gaussian variables with a shared variance and common ROI mean, similar to the model proposed by [7]. However, our model also exploits the simultaneously recorded fMRI data to generate task-dependent ROIs specific to

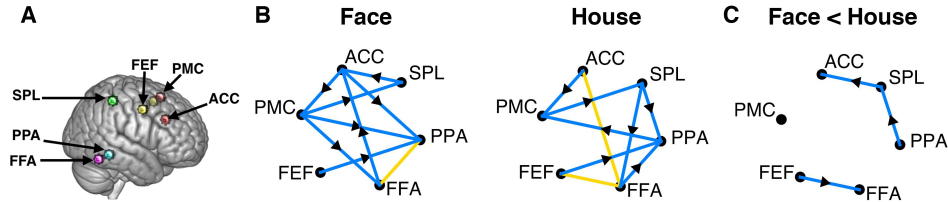

Figure 3: Network connectivity patterns estimated from simultaneous EEG/fMRI data. A, Illustration of ROI locations. B, Mean network connectivity induced by the face and house stimuli. C, Mean difference in directional connections between face and house. Blue line represents unidirectional connection and yellow line represents bidirectional connection.

each individual subject. Since the ROIs identified by fMRI are much smaller and more localized than those defined by an atlas on a standard brain, it was more reasonable to assume that all sources within one ROI have similar activity. Moreover, important ROIs activated in the task were less likely to be neglected when fMRI information was available. Our simulation study further demonstrated that the estimation error largely increased even when a small number of spurious sources were included in each ROI. Together our results show that the high spatial specificity provided by fMRI is critical to ROI based connectivity analysis.

Our model substantially differs from [7] in that it is designed to explain continuously evolving EEG recordings as opposed to epoched EEG responses. Yang et al. [7] modeled the dynamic connectivity on stimulus-locked evoked responses, a reasonable approach when one is interested in the effect specific to a single class of stimulus. On the other hand, our approach allows one to incorporate multiple exogenous covariates either as external or modulatory inputs in the dynamical system so that one can investigate the causal effects of multiple experimental manipulations simultaneously. In essence, the learned latent neural dynamics become a low dimensional representation of the observed EEG dynamics. Our simulation showed that the algorithm could separate the connectivity matrices induced by different stimuli, even when the sign of the intrinsic and modulatory connectivity was opposite to each other. Since the number of parameters is large in this case, we used sparsity regularization via ARD prior to yield more interpretable results. Nevertheless, our model can also be easily modified to analyze connectivity for resting state experiments. Furthermore, since it is often difficult to acquire large samples of simultaneous EEG-fMRI data and the interpretability of the model is important, we chose a biophysically informed linear EEG forward model as opposed to a deep-learning based approach [23, 24].

One limitation of our model is that we assumed the state noise covariance $\mathbf{Q}_s$ to be diagonal. This is often not true in practice. We also assumed that the sources in the same ROI are independent Gaussian distributed, i.e. $\mathbf{Q}_x$ is diagonal. But these sources could be both spatially and temporally dependent. With the available fMRI information, we can potentially design a more complex spatiotemporal structure for $\mathbf{Q}_x$. In this work, we chose to optimize $\mathbf{R}$ directly. An alternative approach is to keep $\mathbf{Q}_y$ fixed and only optimize with respect to $\mathbf{Q}_x$. We did not compare the difference between the two approaches, but optimizing over $\mathbf{R}$ can be easily done via conjugacy.

Since our model solves the ill-posed EEG inverse problem implicitly, we used information from fMRI as spatial prior to solve the EEG source localization using MNE, and obtained a reasonable initial guess for our algorithm. Another limitation is that we assumed a fixed dipole orientation in the lead field matrix and this orientation was estimated based on MNE. In future work, we plan to treat the dipole orientation as unknown parameter over which to optimize. Finally, the temporally continuous nature of our estimation scheme provides an easy framework to incorporate fMRI time series at each ROI so that temporal information from both EEG and fMRI can be used to infer the latent neural dynamics. Our future work will investigate different choices of generative models for fMRI signals to better integrate with EEG.

**Acknowledgement**

This work was supported by the Army Research Laboratory (Cooperative Agreement W911NF-10-2-0022) and the Army Research Office (Grant W911NF-16-1-0507).

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
