[Supplementary Material]

# Appendix

## Derivation of the VB algorithm

### VB-E step

In this step, the posterior distribution of latent variables $q(\mathbf{S}|\mathbf{Y})$ is updated sequentially at every time $t$ via Kalman filtering and smoothing, given the current estimate of $q(\theta|\mathbf{Y})$. Rewrite our state space model in a compact form as:

$$\mathbf{s}_t = \mathbf{G}_t\mathbf{s}_{t-1} + \mathbf{D}\mathbf{u}_t + \boldsymbol{\omega}_t$$
$$\mathbf{y}_t = \mathbf{C}\mathbf{s}_t + \boldsymbol{\phi}_t \tag{1}$$

where $\mathbf{G}_t = \mathbf{A} + \sum_{k=1}^{K} m_t^k \mathbf{B}^k$.

**Kalman filtering**   In the filtering step, we compute the posterior mean $\boldsymbol{\mu}_t^t$ and posterior covariance $\boldsymbol{\Sigma}_t^t$ of latent variables $\mathbf{s}_t$ at time step $t$ given the observations at $t = 1, 2, ...t$, i.e.

$$p(\mathbf{s}_t|\mathbf{y}_1, \mathbf{y}_2, ..., \mathbf{y}_t) = N(\boldsymbol{\mu}_t^t, \boldsymbol{\Sigma}_t^t) \tag{2}$$

The forward recursion updates for $\boldsymbol{\mu}_t^t$ and $\boldsymbol{\Sigma}_t^t$ at $t = 1, 2, ..., T$ are given by

$$\boldsymbol{\mu}_t^t = \boldsymbol{\mu}_t^{t-1} + \mathbf{K}_t(\mathbf{y}_t - \mathbf{C}\boldsymbol{\mu}_t^{t-1})$$
$$\boldsymbol{\mu}_t^{t-1} = \mathbf{G}_t\boldsymbol{\mu}_{t-1}^{t-1} + \mathbf{D}\mathbf{u}_t$$
$$\boldsymbol{\Sigma}_t^t = \boldsymbol{\Sigma}_t^{t-1} - \mathbf{K}_t\mathbf{C}\boldsymbol{\Sigma}_t^{t-1}$$
$$\boldsymbol{\Sigma}_t^{t-1} = \mathbf{G}_t\boldsymbol{\Sigma}_{t-1}^{t-1}\mathbf{G}_t' + \mathbf{Q}_s \tag{3}$$

The Kalman gain is given by

$$\mathbf{K}_t = \boldsymbol{\Sigma}_t^{t-1}\mathbf{C}'(\mathbf{R} + \mathbf{C}\boldsymbol{\Sigma}_t^{t-1}\mathbf{C}')^{-1} \tag{4}$$

The forward recursion is initialized with $\boldsymbol{\mu}_0^0 = \mathbf{s}_0$ and $\boldsymbol{\Sigma}_0^0 = \mathbf{Q}_0$, where we set $\mathbf{s}_0 = \mathbf{0}$ and $\mathbf{Q}_0 = 0.01\mathbf{I}_S$.

**Kalman smoothing**   In the smoothing step, we compute the posterior mean $\boldsymbol{\mu}_t^T$ and posterior covariance $\boldsymbol{\Sigma}_t^T$ of latent variables $\mathbf{s}_t$ at time step $t$ given all observations at $t = 1, 2, ...T$, i.e.

$$p(\mathbf{s}_t|\mathbf{y}_1, \mathbf{y}_2, ..., \mathbf{y}_T) = N(\boldsymbol{\mu}_t^T, \boldsymbol{\Sigma}_t^T) \tag{5}$$

The backward recursion updates for $\boldsymbol{\mu}_t^T$ and $\boldsymbol{\Sigma}_t^T$ at $t = T-1, T-2, ..., 1$ are given by

$$\boldsymbol{\mu}_t^T = \boldsymbol{\mu}_t^t + \mathbf{V}_t(\boldsymbol{\mu}_{t+1}^T - \mathbf{G}_{t+1}\boldsymbol{\mu}_t^t - \mathbf{D}\mathbf{u}_{t+1})$$
$$\boldsymbol{\Sigma}_t^T = \boldsymbol{\Sigma}_t^t + \mathbf{V}_t(\boldsymbol{\Sigma}_{t+1}^T - \boldsymbol{\Sigma}_{t+1}^t)\mathbf{V}_t' \tag{6}$$

where $\mathbf{V}_t$ is defined as

$$\mathbf{V}_t = \boldsymbol{\Sigma}_t^t\mathbf{G}_{t+1}'(\boldsymbol{\Sigma}_{t+1}^t)^{-1} \tag{7}$$

The backward recursion is initialized with $\boldsymbol{\mu}_T^T$ and $\boldsymbol{\Sigma}_T^T$ from the forward recursion step. Note that $\boldsymbol{\Sigma}_{t+1}^t$, $\boldsymbol{\mu}_t^t$ and $\boldsymbol{\Sigma}_t^t$ have already been computed in the forward recursion step.

The following expectations will also be needed in the VB-M step.

$$\mathbf{P}_t = \mathbb{E}[\mathbf{s}_t\mathbf{s}_t'] = \boldsymbol{\Sigma}_t^T + \boldsymbol{\mu}_t^T\boldsymbol{\mu}_t^{T'}$$
$$\mathbf{P}_{t,t-1} = \mathbb{E}[\mathbf{s}_t\mathbf{s}_{t-1}'] = \mathbf{V}_{t-1}\boldsymbol{\Sigma}_t^T + \boldsymbol{\mu}_t^T\boldsymbol{\mu}_{t-1}^{T'} \tag{8}$$

### VB-M step

In this step, we derive closed-form posterior updates of $q(\boldsymbol{\theta}|\mathbf{Y})$ using conjugate priors, given the current estimate of $q(\mathbf{S}|\mathbf{Y})$. The solution is given by [1]:

$$\log q(\boldsymbol{\theta}|\mathbf{Y}) \propto \mathbb{E}_{\mathbf{S}}(\log p(\mathbf{S}, \boldsymbol{\theta}, \mathbf{Y})) \tag{9}$$

Based on the conditional independence in the graphical model, the posterior distribution $q(\boldsymbol{\theta}|\mathbf{Y})$ can be further factorized into

$$q(\boldsymbol{\theta}|\mathbf{Y}) = q(\boldsymbol{\theta}^S|\mathbf{Y})q(\mathbf{R}|\mathbf{Y}) \tag{10}$$

where $\boldsymbol{\theta}^S = \left\{\mathbf{A}, \{\mathbf{B}_k\}_{k=1}^K, \mathbf{D}, \mathbf{Q}_s\right\}$.

**Posteriors of $\boldsymbol{\theta}^S$** Since we assume the state noise covariance $\mathbf{Q}_s$ to be diagonal, we can estimate each row in the model parameters $\mathbf{A}, \{\mathbf{B}_k\}_{k=1}^K, \mathbf{D}, \mathbf{Q}_s$ separately. Specifically, the $r^{th}$ row of the state equation can be expressed as:

$$s_t[r] = \left( \mathbf{a}[r] + \sum_{k=1}^K \mathbf{b}^k[r] m_t^k \right) \mathbf{s}_{t-1} + d[r] u_t[r] + \omega_t[r] \tag{11}$$

$$\omega_t[r] \sim \mathcal{N}(0, \beta^{-1}[r])$$

where $\beta[r] = \frac{1}{\mathbf{Q}_s(r,r)}$ is a scalar denoting the precision of the state noise at the $r^{th}$ row; $\mathbf{a}[r]$ and $\mathbf{b}^k[r]$ are the $r^{th}$ rows of $\mathbf{A}$ and $\mathbf{B}_k$, respectively; $d[r]$ is the $r^{th}$ diagonal element of $\mathbf{D}$. We collect the $r^{th}$ row of the model parameters in the state equation as an $(S + KS + 1) \times 1$ vector $\boldsymbol{\eta}[r] = [\mathbf{a}[r], \mathbf{b}_1[r], ..., \mathbf{b}_K[r], d[r]]'$ and reformulate the $r^{th}$ row of the state equation as:

$$s_t[r] = \boldsymbol{\eta}'[r] \begin{bmatrix} \tilde{\mathbf{F}}_t \mathbf{s}_{-1} \\ u_t[r] \end{bmatrix} + \omega_t[r] = \boldsymbol{\eta}'[r] \tilde{\mathbf{s}}_t[r] + \omega_t[r] \tag{12}$$

where $\tilde{\mathbf{F}}_t = \begin{bmatrix} \mathbf{I}_S & m_t^1 \mathbf{I}_S \dots & m_t^K \mathbf{I}_S \end{bmatrix}'$ and $\tilde{\mathbf{s}}_t[r] = \begin{bmatrix} \tilde{\mathbf{F}}_t \mathbf{s}_{-1} \\ u_t[r] \end{bmatrix}$.

We assume the following Gaussian-Gamma conjugate priors for $\boldsymbol{\eta}[r]$ and $\beta[r]$ [2]:

$$p(\boldsymbol{\eta}[r], \beta[r] | \boldsymbol{\alpha}) = \mathcal{N}\left(0, (\beta[r] \boldsymbol{\Lambda}_{\boldsymbol{\alpha}})^{-1}\right) \text{Gamma}(a_0, b_0) \tag{13}$$

where $\boldsymbol{\alpha} = [\alpha_1, \alpha_2, ..., \alpha_{(K+1)S+1}]$ is a vector of hyperparameters on each element of $\boldsymbol{\eta}[r]$ and $\boldsymbol{\Lambda}_{\boldsymbol{\alpha}}$ is a diagonal matrix with the vector $\boldsymbol{\alpha}$. We also choose a separate Gamma prior for each hyperparameter in $\boldsymbol{\alpha}$ as:

$$p(\boldsymbol{\alpha}) = \prod_{i=1}^{(K+1)S+1} \text{Gamma}(c_0, d_0) \tag{14}$$

Applying Equation (9),

$$\log q(\boldsymbol{\eta}[r], \beta[r] | \mathbf{Y}) \propto \mathbb{E}_{\mathbf{s}, \boldsymbol{\alpha}} \left[ \log p(\mathbf{s} | \boldsymbol{\eta}, \beta) p(\boldsymbol{\eta} | \beta, \boldsymbol{\alpha}) p(\boldsymbol{\alpha}) p(\beta) \right] \tag{15}$$

$$\log q(\boldsymbol{\alpha} | \mathbf{Y}) \propto \mathbb{E}_{\mathbf{s}, \boldsymbol{\eta}[r], \beta[r]} \left[ \log p(\boldsymbol{\eta} | \beta, \boldsymbol{\alpha}) p(\boldsymbol{\alpha}) p(\beta) \right] \tag{16}$$

The variational joint posterior for $\boldsymbol{\eta}[r]$ and $\beta[r]$ has the same form as their priors:

$$q(\boldsymbol{\eta}[r], \beta[r] | \mathbf{Y}) = \mathcal{N}(\bar{\boldsymbol{\mu}}[r], \beta^{-1}[r] \bar{\boldsymbol{\Sigma}}[r]) \text{Gamma}(\bar{a}[r], \bar{b}[r]) \tag{17}$$

where

$$\bar{\boldsymbol{\Sigma}}^{-1}[r] = \begin{bmatrix} \sum_{t=2}^T \tilde{\mathbf{F}}_t \mathbf{P}_{t-1} \tilde{\mathbf{F}}_t' & \sum_{t=2}^T \tilde{\mathbf{F}}_t \boldsymbol{\mu}_{t-1}^T u_t[r] \\ \sum_{t=2}^T u_t[r] (\boldsymbol{\mu}_{t-1}^T)' \tilde{\mathbf{F}}_t' & \sum_{t=2}^T (u_t[r])^2 \end{bmatrix} + \mathbb{E}_{\boldsymbol{\alpha}}(\boldsymbol{\Lambda}_{\boldsymbol{\alpha}}) \tag{18}$$

$$\bar{\boldsymbol{\mu}}[r] = \bar{\boldsymbol{\Sigma}}[r] \begin{bmatrix} \sum_{t=2}^T \tilde{\mathbf{F}}_t \mathbb{E}_{\mathbf{s}}[s_t[r] \mathbf{s}_{t-1}] \\ \sum_{t=2}^T u_t[r] \mu_t^T[r] \end{bmatrix} \tag{19}$$

$$\mathbb{E}_{\boldsymbol{\alpha}}(\Lambda_{\alpha}) = \text{diag}\left( \frac{\bar{c}_1}{\bar{d}_1}, \frac{\bar{c}_2}{\bar{d}_2}, ..., \frac{\bar{c}_{(K+1)S+1}}{\bar{d}_{(K+1)S+1}} \right) \tag{20}$$

$$\bar{a}[r] = a_0 + \frac{T-1}{2} \tag{21}$$

$$\bar{b}[r] = b_0 + \frac{1}{2} \left[ \sum_{t=2}^T \mathbb{E}_{\mathbf{s}}[(s_t[r])^2] - \bar{\boldsymbol{\mu}}'[r] \bar{\boldsymbol{\Sigma}}^{-1}[r] \bar{\boldsymbol{\mu}}[r] \right] \tag{22}$$

$\mathbb{E}_{\mathbf{s}}[s_t[r] \mathbf{s}_{t-1}]$ is the transpose of the $r^{th}$ row of $\mathbf{P}_{t,t-1}$ and $\mathbb{E}_{\mathbf{s}}[(s_t[r])^2]$ is the $r^{th}$ diagonal element of $\mathbf{P}_t$.

The posterior for each hyperparameter $\alpha_j, j = 1, 2, ..., (K+1)S+1$ can be computed independently:

$$q(\alpha_j | \mathbf{Y}) = \text{Gamma}(\alpha_j | \bar{c}_j, \bar{d}_j) \tag{23}$$

where

$$\bar{c}_j = c_0 + \frac{1}{2} \tag{24}$$

$$\bar{d}_j = d_0 + \frac{1}{2}\left[\frac{\bar{a}[r]}{\bar{b}[r]}(\bar{\boldsymbol{\mu}}[r,j])^2 + \bar{\boldsymbol{\Sigma}}_r[j,j]\right] \tag{25}$$

$\bar{\boldsymbol{\mu}}[r,j]$ is the $j^{th}$ element of $\bar{\boldsymbol{\mu}}[r]$ and $\bar{\boldsymbol{\Sigma}}_r[j,j]$ is the $j^{th}$ diagonal element of $\bar{\boldsymbol{\Sigma}}[r]$.

The variational posteriors for $\boldsymbol{\eta}[r], \beta[r]$, and $\boldsymbol{\alpha}$ are computed for $r = 1, 2, ..., S$ separately, from which we obtain the posteriors of state parameters $\boldsymbol{\theta}^S = \{\mathbf{A}, \{\mathbf{B}_k\}_{k=1}^K, \mathbf{D}, \mathbf{Q}_s\}$.

**Posterior of R** The noise covariance $\mathbf{R} = \mathbf{Q}_y + \mathbf{L}\mathbf{Q}_x\mathbf{L}'$ has two unknown covariance matrices $\mathbf{Q}_x$ and $\mathbf{Q}_y$. Since it is difficult to choose a conjugate prior for $\mathbf{Q}_x$ and $\mathbf{Q}_y$ separately, we set a conjugate prior on $\mathbf{R}$ directly. Applying Equation (14):

$$\log q(\mathbf{R}|\mathbf{y}) \propto \mathbb{E}_{\mathbf{s}}(\log p(\mathbf{y}|\mathbf{s},\mathbf{R})p(\mathbf{R})) \tag{26}$$

We set the inverse Wishart prior $IW(v_0, \mathbf{V}_0)$ on $\mathbf{R}$ [3]:

$$p(\mathbf{R}) = \frac{|\mathbf{V}_0|^{\frac{v_0}{2}}}{2^{\frac{v_0 M}{2}}\Gamma_M(\frac{v_0}{2})}|\mathbf{R}|^{-\frac{v_0+M+1}{2}}\exp\left(-\frac{1}{2}\mathrm{Tr}(\mathbf{V}_0(\mathbf{R})^{-1})\right) \tag{27}$$

The posterior is given by:

$$q(\mathbf{R}|\mathbf{y}) = IW(v_n, \mathbf{V}_n) \tag{28}$$

where

$$v_n = v_0 + T \tag{29}$$

$$\mathbf{V}_n = \mathbf{V}_0 + \left(\sum_{t=1}^T (\mathbf{y}_t - \mathbf{C}\boldsymbol{\mu}_t^T)(\mathbf{y}_t - \mathbf{C}\boldsymbol{\mu}_t^T)' + \mathbf{C}\boldsymbol{\Sigma}_t^T\mathbf{C}'\right) \tag{30}$$

We have $\mathbb{E}(\mathbf{R}) = \frac{\mathbf{V}_n}{v_n-M-1}$ and $\mathbb{E}(\mathbf{R}^{-1}) = \mathbf{V}_n^{-1}v_n$.

**Computation for ELBO**

The ELBO can be computed by:

$$\begin{aligned}
\mathcal{L}(q) = \sum_{r=1}^S \{&-\frac{1}{2}\sum_{t=2}^T \frac{\bar{a}[r]}{\bar{b}[r]}\left(\mathbb{E}(s_t^2[r]) - 2\mathbb{E}(s_t[r]\bar{\boldsymbol{\mu}}[r]'\tilde{\mathbf{s}}_t) + \mathrm{Tr}\left(\left(\bar{\boldsymbol{\mu}}[r]\bar{\boldsymbol{\mu}}[r]' + \frac{\bar{b}[r]}{\bar{a}[r]}\bar{\boldsymbol{\Sigma}}[r]\right)\mathbb{E}(\tilde{\mathbf{s}}_t\tilde{\mathbf{s}}_t')\right)\right) \\
&-\frac{T-1}{2}\log 2\pi + \frac{1}{2}\log\det(\bar{\boldsymbol{\Sigma}}_{\eta[r]}) + \frac{(K+1)S+1}{2} - \log\Gamma(a_0) + a_0\log b_0 - b_0\frac{\bar{a}[r]}{\bar{b}[r]} \\
&+ \log\Gamma(\bar{a}[r]) - \bar{a}[r]\log\bar{b}[r] + \bar{a}[r] \\
&+ \sum_{j=1}^{\frac{(K+1)S+1}{2}}\left(-\log\Gamma(c_0) + c_0\log d_0 + \log\Gamma(\bar{c}_j[r]) - \bar{c}_j[r]\log\bar{d}_j[r]\right)\} \\
&+ \frac{ST}{2}\log 2\pi + \frac{1}{2}\sum_{t=1}^T \log\det(\boldsymbol{\Sigma}_t^T) + \frac{ST}{2} \\
&- \frac{T}{2}\log 2\pi - \frac{T}{2}\log\det\mathbf{R} \\
&- \frac{1}{2}\mathrm{Tr}\left[(\mathbf{R})^{-1}\left(\sum_{t=1}^T(\mathbf{y}_t - \mathbf{C}\boldsymbol{\mu}_t^T)(\mathbf{y}_t - \mathbf{C}\boldsymbol{\mu}_t^T)' + \mathbf{C}\boldsymbol{\Sigma}_t^T\mathbf{C}'\right)\right]
\end{aligned} \tag{31}$$

**Additional information on simulations**

The lead field matrix $\boldsymbol{L}$ used in the simulation was computed based on the anatomical MRI data of a single subject using the FieldTrip toolbox [7]. We first generated a Boundary Element Method (BEM) volume conductor model based on the head geometry obtained from the T1-weighted structural MR image of the subject. The head model specifies the conductivity of three different tissue types: brain, skull and scalp. We then constructed a 3-D volumetric source model that discretizes the brain volume using a uniformly spaced 3-D grid with 5 mm spatial resolution. It contains 24948 source points (dipoles) in total. The spatial placement of 34 EEG electrodes was manually aligned to the same coordination system as the head model and the source model. The lead field matrix was computed for each source point using a fixed dipole orientation estimated from the EEG data using MNE. Hence, a lead field matrix $\boldsymbol{L}$ with dimension $34 \times 24948$ was obtained. We selected 5 ROIs based on the fMRI activation map (see section 3.2) specific to the subject at the face fusiform area (FFA), parahippocampal place area (PPA), superior parietal lobule (SPL), anterior cingulate cortex (ACC), and frontal eye field (FEF). In total, 117 true sources were included across all 5 ROIs (FFA: 32, PPA: 24, SPL: 9, ACC: 33, FEF: 19). Therefore there are 117 rows in the binary indicator matrix $\mathbf{G}$ that are not all zeros. When fMRI data is not available, one can only select ROIs based on atlases defined in a standard space, which results in inaccurate localization of ROIs for each individual subject. To simulate the loss in spatial specificity, we dilated the ROIs such that only a small number of spurious sources (38 in total) were added to each ROI (FFA: 10, PPA: 7, SPL: 4, ACC: 15, FEF: 2). We compared the performance of the algorithm in this "EEG-only" condition with the "EEG-fMRI" condition in which the spatial localization of ROIs is accurate. The only difference between the two conditions was the number of nonzero rows in the indicator matrix $\mathbf{G}$. Figure 1 shows the network configuration used in the simulation. Scenarios 1 and 2 have the same external input $u$ and intrinsic connectivity matrix $\mathbf{A}$, but they differ in the number and type of modulatory inputs. $\mathbf{m}_1$ in scenario 1 represents a block-design while $\mathbf{m}_2$ and $\mathbf{m}_3$ in scenario 2 represent an event-related design. Scenario 2 was designed to mimic our real data experiment. Figure 2 and Figure 3 show an example of the comparison between "EEG-fMRI" and "EEG-only" conditions in recovering the connectivity matrices and noise covariance matrices, for scenarios 1 and 2 respectively. In both cases, the "EEG-fMRI" condition yields smaller error and less false positive connections. In practice, it is common to get inaccurate ROIs or even neglect important ROIs with EEG information alone. Our simulations illustrate the value of simultaneous EEG-fMRI data.

Figure 1: Network configuration in the simulation.

Figure 2: Comparison of posterior estimates between "EEG-fMRI" and "EEG-only" in scenario 1. Connections in $\mathbf{A}$ and $\mathbf{B1}$ were thresholded at $P < 0.05$ after Bonferroni correction.

**Additional information on algorithm initialization and runtime**

For both simulation scenarios, we initialized the algorithm in the same way. The algorithm was initialized by solving the least squares in Equation (1) to obtain estimates of $\hat{\mathbf{s}}_t$, $\hat{\mathbf{x}}_t$, $\hat{\mathbf{Q}}_x$, $\hat{\mathbf{Q}}_s$, $\hat{\mathbf{A}}$, $\hat{\mathbf{B}}^{\mathbf{k}}$, and $\hat{\mathbf{D}}$. $\hat{\mathbf{Q}}_{\mathbf{y}}$ was estimated from the data during baseline period. Small non-informative priors were chosen for the model parameters such that the posterior largely depends on the data likelihood. For state noise precision, we set $a_0 = 10^{-4}, b_0 = 10^{-3}$. For hyper-parameter $\mathbf{\Lambda}_{\boldsymbol{\alpha}}$, we set $c_0 = 10^{-2}, d_0 = 10^{-9}$. Choosing $d_0 \ll c_0$ increases the sparsity penalty. For covariance $\mathbf{R}$, we set $v_0 = 35$. In Figure 4, we showed the relative error between the final posterior estimates and their initial values. In both scenarios, the final posterior estimates changed substantially from their initial values and moved closer towards the true values. The relative error between the posterior estimates and the true values is significantly smaller for all model parameters. The runtime of the Matlab implementation of the algorithm on one set of the data in simulation 1 (comparable to real data) is 935 s on a 2.8 GHz Intel Core i7 Mac machine.

Figure 3: Comparison of posterior estimates between "EEG-fMRI" and "EEG-only" in scenario 2. Connections in **A**, **B3**, and **B3** were thresholded at $P < 0.05$ after Bonferroni correction.

Figure 4: Comparison between the posterior estimates and their initial values. A, Relative error of the initial values vs. posterior estimates and true values vs. posterior estimates in scenario 1. B, Similar comparison in scenario 2. Error bar represents the standard error of the mean across 10 independent simulations.

**Additional information on simultaneous EEG-fMRI experiment**

Simultaneous EEG and fMRI were recorded when subjects performed an event-related three-choice visual categorization task. On each trial, an image of a face, car, or house was presented for 100 ms. Subjects reported their choice of the image category by pressing one of the three buttons on an MR-compatible button response pad. The stimuli consisted of a set of 30 face, 30 car, and 30 house images. The phase coherence of the images was degraded at a high coherence (50%) level and at a low coherence (35%) level by a weighted mean phase algorithm [4]. The phase coherence

modulates the amount of sensory evidence in the stimuli and thus influences the decision ambiguity. Each subject completed 4 runs of the categorization task. In each run, there were 180 trials (30 per condition; 6 conditions: face high, car high, house high, face low, car low, and house low). EEG data were recorded simultaneously with the fMRI data at 1 kHz sampling rate using a custom-built MR-compatible EEG system with differential amplifiers and bipolar EEG montage. The caps were configured with 36 Ag/AgCl electrodes, including left and right mastoids, arranged as 43 bipolar pairs. More details in data recording and experiment design can be found in [5, 6]. The repetition time (TR) of fMRI is 2 s.

In the preprocessing of EEG data, we first removed the gradient artifacts by subtracting from each functional volume an average artifact template obtained from across all functional volume acquisitions. We then smoothed the data with a 10 ms median filter to attenuate any residual spike artifacts. Subsequently, we performed a standard EEG noise removal with a 0.5 Hz high-pass filter to remove direct current drift, 60 and 120 Hz notch filters to remove electrical line noise, and a 100 Hz low-pass filter to remove high-frequency artifacts not associated with neurophysiological processes. Ballistocardiogram (BCG) artifact was then removed by a conservative approach, based on principal component analysis to reduce the risk of signal power loss. These BCG-free data were then rereferenced from the 43 bipolar channels to the 34-electrode space, and then to the common average of all channels. Finally, EEG data were downsampled to 100 Hz. Stimulus-locked EEG epochs with a duration of 1000 ms (500 ms prestimulus to 500 ms poststimulus) were extracted from the BCG-free data for the algorithm initialization. The baseline was chosen from 200 ms prestimulus to stimulus onset and the average voltage during the baseline period was subtracted from the epoch. EEG measurement noise covariance $\hat{\mathbf{Q}}_y$ was then estimated from the baseline period. The free source orientation lead field matrix $\mathbf{L}$ was first computed using the 'BEMCP' head model in FieldTrip Toolbox [7].

For each subject, we identified 6 ROI (FFA, PPA, SPL, ACC, PMC, FEF, see Figure 5A) that showed differential activity for face vs. nonface from an EEG-informed fMRI analysis [5]. FFA and PPA were identified for each subject based on a separate functional localizer task. Consequently, these ROIs were task-specific and varied in size depending on each subject's anatomical geometry. The algorithm was initialized first by solving the EEG inverse problem using MNE, the inverse operator $\mathbf{W}$ is given by [8]:

$$\mathbf{W} = \mathbf{Q}_0 \mathbf{L}'(\mathbf{L}\mathbf{Q}_0\mathbf{L}' + \lambda^2\hat{\mathbf{Q}}_y)^{-1} \tag{32}$$

where $\mathbf{Q}_0$ is the diagonal source covariance matrix. We set the variance of sources at each ROI to 1 and the variance of sources outside any ROI to 0.1. $\lambda$ is a regularization parameter calculated as $\frac{\text{trace}\mathbf{L}\mathbf{Q}_0\mathbf{L}'}{\text{trace}\hat{\mathbf{Q}}_y \cdot \text{SNR}}$. Therefore, the estimated source activity $\hat{\mathbf{x}}_t$ is given by:

$$\hat{\mathbf{x}}_t = \mathbf{W}\mathbf{y}_t \tag{33}$$

For each dipole in $\hat{\mathbf{x}}_t$, we calculated the first singular value across its three components at x, y, and z directions. The first singular vector was used as the dipole orientation to calculate a lead field matrix $\mathbf{L}$ with fixed dipole orientation in our model. $\hat{\mathbf{s}}_t$ was estimated from the mean activity of $\hat{\mathbf{x}}_t$ at each ROI. The initial values of other model parameters were then computed by solving a set of least squares in Equation (1). We modeled face, car, and house stimuli as three separate modulatory inputs. In addition, face stimuli were fed into FFA and house stimuli fed into PPA as external inputs. Car stimuli served as a control condition and we focused on the comparison between faces and houses in this analysis. We fit the state-space model to the continuous noise-cleaned data from each of the 4 runs separately. In addition to comparing the connectivity matrices induced by faces and houses, we also qualitatively checked how well our model fit to the data. Based on the posterior estimates of model parameters, we calculated the model prediction of EEG observations. Since we only modeled three stimulus events and there are many other covariates that contributed to the variance in the observed EEG (such as motor response), we focused on the comparison between the predicted and observed event-related potentials (ERPs). Figure 5B shows the predicted ERP and actual ERP responses at 34 channels of a representative subject, averaged across three stimulus types. The observed ERP responses were well accounted for by the model, for example at occipital electrodes (Figure 5C).

Figure 5: Predicted ERP and Actual ERP responses from one representative subject. A, Illustration of ROI locations in subject's anatomical space. B, Grand average of ERPs (Actual vs. Predicted) across three categories at all 34 channels. Scalp topology was shown at t=235 ms post-stimulus onset. C, Actual vs. Predicted ERPs averaged across three categories at 4 occipital channels. Shaded area represents the standard error of the mean.