[Reviews · NeurIPS 2019]

Reviewer 1



This paper develops a novel method to infer directional relationships between cortical areas of the brain based on simultaneously acquired EEG and fMRI data. Specifically, the fMRI activations are used to select ROIs related to the paradigm of interest. This information is used in a coupled state-space and forward propagation model to identify robust spatial sources and directional connectivity. The authors use a variational Bayesian framework to infer the latent posteriors and noise covariances. They demonstrate the power of joint EEG/fMRI analysis using two simulated experiments and a real-world dataset. Overall, this is a well-developed paper that presents an elegant new framework for directional EEG analysis, informed by an auxiliary data modality. I also appreciate the rigorous evaluation on both synthetic and real-world data. Directional connectivity is an increasingly popular topic of study in computational neuroscience, and the coupled state-space model is one of the more principled frameworks that I have seen. For these reasons, I would recommend this paper for acceptance. With that said, there were a few issues that dampened my enthusiasm for the paper. I would suggest that the authors address these points in a future iteration of this work. === Major Concerns === My main complaint is that even though the paper claims to develop a multimodal framework, the fMRI data is not well-integrated into the generative model. Practically speaking, the fMRI information is used to select the ROIs used in the Bayesian model. This process is done a priori, almost like a preprocessing step. However, fMRI has its own forward model (GLM) which mimics the form of Eq. (2). I would have liked to see the GLM folded into the Bayesian framework. For example, this joint analysis can mitigate cases when the fMRI activations are missing or unreliable by adding noise into the ROI boundaries. My second concern is the relatively small number of ROIs (5-6) assumed in the experimental section. While it may be appropriate for the chosen paradigm, ultimately, we would like to know more about information processing across distributed functional systems, analogous to resting-state data. Given the large number of parameters, it is unclear whether the inference procedure would be robust in the case of a larger state-space. If not, this would greatly limit the generalizability of this work to new applications. === Minor Points === 1) I would have preferred more details about the inference procedure (including update equations) and a much shorter discussion of the experimental setup and results. 2) The model initialization in Section 3.2 is very sophisticated. How do the final values differ from this initialization? Does the model produce reasonable results from a simple random intializaton? 3) The Discussion mentions that the ROIs are specific to each subject. While the spatial location may have been guided by the fMRI, as far as I can tell, all subjects had the same number of ROIs. Can this method be applied when the number of ROIs differs across subjects (as is often the case for fMRI activations)? ** UPDATE: The author response is good, in that it discusses a number of reviewer concerns. It solidifies my opinion that this is a good paper worthy of acceptance. With that said, using the fMRI as a precomputed spatial prior is not particularly innovative as far as multimodal approaches go.

Reviewer 2



This paper proposes a probabilistic model-based method for estimating effective connectivity among brain regions based on simultaneous EEG and fMRI recordings. The model is a linear state-space model for sensor-space EEG data, with the observation (forward) model pre-specified from a given lead-field matrix and task-based ROIs defined by fMRI. The technical idea sounds reasonable, although rather straightforward and the advance from previous studies like [7] appears to be quite small. Most crucially, the method does not seem to really exploit the advantage of simultaneous measurements because the task-based ROIs can actually be defined even from fMRI-only measurement during participants do the same task. The paper itself is very well-written, with high clarity. The simulation and real-data experiment are well conducted, although it is not easy to validate the method from the real-data result. As minor points, is it necessary to assume that Q_x is diagonal? Some comments will be useful on how the technical idea of this paper differs from previous uses of fMRI for spatial priors [12-14]. Comments after authors response: I still think that the method is not far beyond the previous work [7] because the key idea of reducing ROI-based source connectivity estimation into sensor-space AR model fitting has already been presented in that work and other modifications on the model is rather straightforward once the formulation is given. Some specific modeling and algorithmic details (e.g. modulatory connectivity, initialization scheme) are likely new and useful, which I initially overlooked.

Reviewer 3



In this work, the Authors propose a linear state-space model for estimating effective connectivity using EEG and fMRI data. The fMRI data is used to localize the ROIs and constraint the optimization of the model inferred from EEG data. The temporal dependence between latent variables is modeled as a first-order MVAR in the presence of external and context-dependent inputs. Such a model is complemented by a linear forward model for propagation. Variational-Bayes inference is used to estimate the model, adopting some strong assumptions, e.g. Q_s diagonal. Experiments on simulated data are conducted in order to characterize the importance of constraining sources to what fMRI data suggests, both in a block and in an event-related design. On real fMRI/EEG data, from a face, car, and house stimulation paradigm, the proposed method is used to study the connectivity patterns between FFA, PPA, SPL, ACC, FEF and PMC for face-house processing, suggesting that additional connectivity is required for recognizing a house relative to a face. The manuscript is very well written and the proposed method is very interesting and grounded. Issues: - The main issue is the absence of the code implementing the proposed method. The Authors do not commit even to publishing the code of the proposed method and simulations after acceptance. The Authors do not offer any explanation for this decision which, in 2019, is pretty difficult to accept. Same thing for the EEG/fMRI dataset. It is true that many practical details are present in the manuscript and the supplementary materials. Still, reproducibility of the results seems not a major concern for the Authors. But it is for me. - What about the time required by the entire method to be computed? No mention of that.

[Author Response · NeurIPS 2019]

We thank the reviewers for their thoughtful reviews and below we address their major concerns. We first emphasize that
we will publish both the data and code if the paper is accepted—-this was an oversight by us for not making clear we
would do so. As both Reviewers 1 and 2 point out, our current version of the state-state model uses task-specific ROIs
identified from fMRI to inform the modeling of directed connectivity in EEG source space. Although the information
from fMRI is incorporated only as a spatial prior, our simulation results in Section 3.1 demonstrate the importance of
having accurate localization of ROIs (see blue bars for relative errors in Figure 2). It is clear that model performance
decreases markedly even when a small number of sources are erroneously included in the ROIs. While it is possible
to define ROIs from a separate fMRI experiment, one would expect increased variability in the shared neural activity
between two non-simultaneously acquired modalities, leading to less accurate ROI boundaries and/or spurious ROI
regions. This variability would be expected even from different recording sessions for the same subject. As a result,
our model emphasizes the value of collecting simultaneous EEG-fMRI and using the two modalities to exploit their
respective expressive power (spatial localization with fMRI and temporal dynamics in the EEG). Nevertheless, we
agree with the reviewers that one might expect that the current model can be improved if fMRI information was also
symmetrically integrated into the state-space framework.

In fact, in a previous version of this model, we did consider a generative process of fMRI which links to the EEG via the
common latent dynamics $\mathbf{s}_t$. The fMRI BOLD signal was modeled as a linear convolution between the latent variable
and canonical hemodynamic response functions (HRFs), which is expressed in linear matrix product form as:

$$\mathbf{z}_t[r] = [\mathbf{s}_t[r], \mathbf{s}_{t-1}[r], ..., \mathbf{s}_{t-L+1}[r]]', \quad y_t^F[r] = \mathbf{w}'[r]\mathbf{H}\mathbf{z}_t[r] + \phi_t^F[r] \tag{1}$$

where $\mathbf{z}_t[r]$ is a vector of $L$ lagged values of the latent variable at the $r^{th}$ ROI. $\mathbf{H}$ is a $3 \times L$ matrix that denotes a set of
three hemodynamic basis functions, and $\mathbf{w}[r]$ is a $3 \times 1$ weight vector on the hemodynamic basis functions. Here we
estimate a different $\mathbf{w}[r]$ for each ROI to account for the regional hemodynamic response variability. $\phi_t^F[r]$ is i.i.d.
Gaussian noise at the $r^{th}$ ROI. Our simulation results for this model showed, however, that the addition of BOLD
time series **does not** improve the estimation of latent dynamics compared to using EEG time series alone. This is not
surprising since the temporal scale of the BOLD signal is much slower ($\sim 200$ times slower) than that of the latent
dynamics. It is the spatial specificity of the ROI localization offered by simultaneously recorded fMRI that contributes
most to an accurate recovery of the latent dynamics. Another reason why we excluded the fMRI equation in the current
paper is that we found using a linear convolution with canonical HRF functions did not predict the fMRI signals well.
This is probably because the canonical HRF functions are estimated from a deconvolution between the task stimulus
function (sparse events) and BOLD. We found that the BOLD signal is better predicted by the latent states convolved
with an oscillatory-shaped HRF function, which substantially differs from the well-studied canonical form–this is an
interesting finding but one that we still need to confirm and better understand. Since relatively little work has been done
predicting BOLD from continuous EEG source activity, we believe that more investigation is needed before we report
these findings.

Though our model shares some similarity with [7], it substantially generalizes this model and potentially has broader
applications. First, our model is designed to explain the variance in continuously evolving EEG recordings as opposed
to epoched EEG responses, as in [7]. This allows researchers to add multiple covariates (e.g., different experimental
conditions) as external inputs into the dynamical system and investigate their individual influence. Second, our model
can be easily applied to data not having a trial-based structure (e.g., resting-state data). We evaluated the model on
task based EEG-fMRI data rather than resting state data because we originally wanted the model to be tested using
a hypothesis-driven approach and our task EEG-fMRI data had been analyzed thoroughly by our group. In addition,
for the resting-state case, there is no task activation so we potentially lose the ability to highlight the importance of
the spatial specificity from fMRI. Nonetheless we agree with Reviewer 1 that more evaluation should be done using
large-scale resting-state data. Also related to Reviewer 1's comments, it is certainly possible to have different numbers
of ROIs for each subject. We chose the same number for each subject because these ROIs were obtained from a
group-level fMRI activation and thus made group-inference on connectivity easier. Running our model with separate
sets of ROIs per subject is possible, but a separate group inference procedure will be needed when subjects have
different number of ROIs.

Another major point/question raised by the reviewers was the sensitivity of our results to our intialization procedure.
Since variational inference algorithms are generally sensitive to the initialization, we chose a more informative
initialization procedure which uses some fMRI-constrained techniques proposed in [12-14] to achieve a better initial
solution for the EEG source localization. Using a completely random initialization is less likely to produce a good
solution since the ELBO is expected to have many local optima. In our case, the final values substantially change
from the initial values and move closer towards their true values; we will add a figure in the supplemental material
showing this. In general, more detailed information about the initialization and algorithm runtime will be included
in the supplement. **Minor concerns:** In the revision we will try our best to reorganize the material to include a more
detailed description of the inference procedure in the main manuscript. We chose a diagonal $\mathbf{Q}_x$ as an approximation in
the spirit of mean-field variational inference. It is not necessary but it simplifies the inference derivation.

[Meta-Review · NeurIPS 2019]

The paper proposes a generative model for inferring directional EEG connectivity. The approach is sound and the manuscript is well written. The Reviewers agree that the charaterization of the proposed method is well supported by both simulated and real data. I would consider a minor concern the issue that there is no real exploitation of the concurrent fMRI/EEG acquisition since the analysis is designed as two independent steps of ROI estimate (fMRI) and connectivity inference (EEG). We may consider this as an open challenge in the research agenda rather than a serious pitfall of the proposed method. As pointed out by one Reviewer I would consider a more meaningful concern the lack of code.